# Connectional asymmetry of the inferior parietal lobule shapes hemispheric specialization in humans, chimpanzees, and rhesus macaques

Luqi Cheng[1,2,3], Yuanchao Zhang[1], Gang Li[2,3,4], Jiaojian Wang[1,5], Chet Sherwood[6], Gaolang Gong[7,8], Lingzhong Fan[2,3,4,9]*, Tianzi Jiang[1,2,3,4,9]*

[1]Key Laboratory for NeuroInformation of Ministry of Education, School of Life Science and Technology, University of Electronic Science and Technology of China, Chengdu, China; [2]Brainnetome Center, Institute of Automation, Chinese Academy of Sciences, Beijing, China; [3]National Laboratory of Pattern Recognition, Institute of Automation, Chinese Academy of Sciences, Beijing, China; [4]University of Chinese Academy of Sciences, Beijing, China; [5]Center for Language and Brain, Shenzhen Institute of Neuroscience, Shenzhen, China; [6]Department of Anthropology and Center for the Advanced Study of Human Paleobiology, The George Washington University, Washington, United States; [7]State Key Laboratory of Cognitive Neuroscience and Learning & IDG/McGovern Institute for Brain Research, Beijing Normal University, Beijing, China; [8]Beijing Key Laboratory of Brain Imaging and Connectomics, Beijing Normal University, Beijing, China; [9]CAS Center for Excellence in Brain Science and Intelligence Technology, Institute of Automation, Chinese Academy of Sciences, Beijing, China

*For correspondence:
lingzhong.fan@ia.ac.cn (LF);
jiangtz@nlpr.ia.ac.cn (TJ)

**Competing interests:** The authors declare that no competing interests exist.

**Abstract** The inferior parietal lobule (IPL) is one of the most expanded cortical regions in humans relative to other primates. It is also among the most structurally and functionally asymmetric regions in the human cerebral cortex. Whether the structural and connectional asymmetries of IPL subdivisions differ across primate species and how this relates to functional asymmetries remain unclear. We identified IPL subregions that exhibited positive allometric in both hemispheres, scaling across rhesus macaque monkeys, chimpanzees, and humans. The patterns of IPL subregions asymmetry were similar in chimpanzees and humans, but no IPL asymmetries were evident in macaques. Among the comparative sample of primates, humans showed the most widespread asymmetric connections in the frontal, parietal, and temporal cortices, constituting leftward asymmetric networks that may provide an anatomical basis for language and tool use. Unique human asymmetric connectivity between the IPL and primary motor cortex might be related to handedness. These findings suggest that structural and connectional asymmetries may underlie hemispheric specialization of the human brain.

## Introduction

The association cortex has expanded greatly in size and exhibits modified connectivity patterns in human brain evolution (*Orban et al., 2006*; *Mars et al., 2017*; *Ardesch et al., 2019*; *Van Essen et al., 2019*). Compared with the primary sensory and motor cortical regions, the association cortex displays disproportionate expansion in conjunction with overall neocortical volume enlargement across primates (*Chaplin et al., 2013*). Accordingly, association areas comprise a large percentage

of the neocortex in human brains (*Orban et al., 2006*; *Van Essen and Dierker, 2007*; *Donahue et al., 2018*). Functional and neuroanatomical asymmetries are also pronounced in the human brain, appearing to be more extreme compared with other primate species, especially in the association cortex (*Chance and Crow, 2007*). Nevertheless, cerebral asymmetry exists not only in humans but also in nonhuman primates (*Gómez-Robles et al., 1761*; *Hopkins, 2013*). For example, olive baboons and chimpanzees showed population-level leftward volumetric asymmetry in the planum temporale, which is thought to be homologous to part of Wernicke's area in humans and may have played a facilitating role in the evolution of spoken language (*Spocter et al., 2010*; *Marie et al., 2018*). Comparative studies on brain asymmetry are crucial for understanding the evolution and function of the modern human brain.

Language and complex tool use, which show considerable lateralization in the human brain, are considered to be universal features of humans (*Johnson-Frey et al., 2005*; *Lewis, 2006*; *Binder et al., 2009*). These specialized functions all involve the inferior parietal lobule (IPL), an area of the association cortex that represents a zone of topographical convergence in the brain (*Johnson-Frey, 2004*; *Binder et al., 2009*). Moreover, the IPL is one of the most expanded regions in humans compared with nonhuman primates (*Orban et al., 2006*; *Van Essen and Dierker, 2007*; *Kaas, 2012*; *Ardesch et al., 2019*). The functional diversity and expansion of the IPL imply that it contains subdivisions that may have been elaborated or developed in the ancestors of modern humans, allowing new abilities such as extensive tool use and communication using gestures (*Kaas, 2012*). However, due to the scarcity of data, different criteria, and methodological limitations for defining regions or subregions (*Mars et al., 2017*), whether the internal organization of the IPL differs across species and how this relates to different asymmetric functions remain unclear.

A major challenge for neuroscience is to translate results obtained using one method and in one species to other methods and other species. Although the IPL has been subdivided into distinct subregions using cytoarchitecture and this technique has provided invaluable information, cellular microstructure alone is insufficient to completely represent brain organization, especially long-range connections, which are the major determinant of regional specialization (*Passingham et al., 2002*; *Caspers et al., 2006*). Furthermore, histological methods with postmortem brains cannot be readily scaled to large populations. Recent advances in diffusion magnetic resonance imaging (MRI), which allow the quantitative mapping of whole-brain neural connectivity in vivo, provide an alternative technique called connectivity-based parcellation to subdivide specific regions of the brain or even the entire cortex (*Fan et al., 2016*; *Eickhoff et al., 2018*). In previous studies, this technique was successfully used to characterize IPL subdivisions in different species as well as to perform cross-species comparisons (*Mars et al., 2011*; *Wang et al., 2020*).

Previous studies have assessed asymmetries of the IPL using local characteristics, such as cortical volume, thickness, and surface area (*Croxson et al., 2018*; *Kong et al., 2018*). However, although such regional asymmetries have been identified, additional analyses are needed to address the architecture of neural connectivity (*Ocklenburg et al., 2016*). A recent 'connectomic hypothesis for the hominization of the brain' suggests neural network organization as an intermediate anatomical and functional phenotype between the genome and cognitive capacities, which are extensively modified in the human brain (*Changeux et al., 2020*). The functions and interactions of brain regions are determined by their anatomical connections (*Passingham et al., 2002*). Therefore, identifying connectional asymmetries may provide new insights into the structural and functional specializations of the human brain.

This study investigated asymmetries of IPL subregions in terms of both structure and anatomical connectivity in rhesus macaques, chimpanzees, and humans. We first used connectivity-based parcellations to subdivide the IPL to reveal consistent cross-species topographical organization. We then investigated the volumetric allometric scaling and asymmetries of the IPL subregions across species. Using vertex-, region of interest (ROI)-, and tract-wise analyses, we examined asymmetries of the IPL subregions in terms of their connectivity profiles and subcortical white matter pathways to identify evolutionary changes.

## Results

### Connectivity-based parcellation

For each species, a data-driven connectivity-based parcellation was applied to group the vertices in the IPL into functionally distinct clusters based on anatomical connectivity (*Figure 1*). Because spectral clustering does not require a specific number of clusters, we iterated the number of subregions from 2 to 12 to search for the optimal number of subregions. To accomplish this, we identified the optimal number of subregions of the IPL by choosing the maximum number of subregions that showed a coherent topological organization across all species while balancing that by the minimum number of subregions that could be identified based on their cytoarchitectural definitions in macaques, chimpanzees, and humans (*Pandya and Seltzer, 1982*; *Reyes, 2017*). The two- to five-cluster solutions are shown in *Figure 1—figure supplement 1*. The two- to four-cluster solutions showed a consistent rostral–caudal pattern in all three species, but in the five-cluster solution, a ventral cluster emerged in chimpanzees and a dorsal cluster emerged in humans. The four-cluster solution revealed a rostral–caudal topological pattern that was consistent with previous parcellations based on cytoarchitecture and anatomical connectivity (*Pandya and Seltzer, 1982*; *Caspers et al., 2006*; *Mars et al., 2011*; *Fan et al., 2016*). Also, the cytoarchitectural definition of macaques revealed four subregions in the IPL (*Pandya and Seltzer, 1982*), which was fewer than the seven cytoarchitectural subregions of the human IPL (*Caspers et al., 2006*). Although the four-cluster solution was not the finest, especially in humans, it contained potentially valuable information about the differences between species. Furthermore, the aim of our research was not to find the 'best' cluster solution for the IPL but to identify an appropriate parcellation that could shed light on the lateralization of the structure and connectivity of the IPL and its subregions in this particular sample of three primate species. As such, we chose four clusters as the optimal solution for the cross-species comparison.

It is widely accepted that the IPL contains two major cytoarchitectural divisions across species, the anterior (PF) and posterior (PG) areas (*von Economo and Koskinas, 1925*; *Von Bonin and Bailey, 1947*; *Bailey et al., 1950*). Our results were consistent with this two-way parcellation and refined it into four subdivisions, specifically, two anterior clusters (the C1 and C2) in the PF and two posterior clusters (the C3 and C4) in the PG. In macaques and chimpanzees, the IPL was previously parcellated into four distinct areas based on histology (*Pandya and Seltzer, 1982*; *Reyes, 2017*) in keeping with our four-cluster solution. In humans, the IPL was cytoarchitecturally parcellated into seven distinct areas. Although we proposed a four-cluster solution that has fewer areas than the cytoarchitectural map, it is also consistent with it (*Caspers et al., 2006*). Specifically, the rostral anterior cluster (C1) is similar to the PFt and part of the PFop area defined using cytoarchitecture by *Caspers et al., 2006*, the caudal anterior cluster (C2) corresponds to the PF and PFm areas, the rostral posterior cluster (C3) is similar to the PGa area, and the caudal posterior cluster (C4) is similar to the PGp area. Our results did not include the PFcm area because it is located deep in the parietal operculum. Given the limited descriptions of subdivisions and connectivity of the IPL in chimpanzees, our parcellation of the IPL can depict the subregions and connectivity of the IPL in chimpanzees from an evolutionary perspective.

To assess which hemisphere was dominant with respect to a given function of the human IPL subregions, we decoded the functions of the human IPL subregions from the Neurosynth database (*Yarkoni et al., 2011*) and calculated differences in the correlation values between the left and right corresponding subregions (*Figure 1—figure supplement 2*). The term *tool* showed a much higher correlation with the left C1 than with the right C1, suggesting that the left C1 is more involved in tool use. Terms such as *tool* and *semantics* showed relatively high correlations with the left C2, whereas terms such as *nogo* and *inhibition* showed relatively high correlations with the right C2, suggesting that the left C2 is more involved in tool use and language, whereas the right C2 is more involved in executive function. Terms such as *retrieval*, *episodic*, *recollection*, *memories*, and *coherent* showed relatively high correlations with the left C3, whereas terms such as *nogo*, *inhibition*, and *beliefs* were correlated with the right C3, suggesting that the left C3 is more involved in memory and language, whereas the right C3 is more involved in executive and social cognitive functions. Terms such as *episodic* and *coherent* showed relatively high correlations with the left C4, whereas terms such as *spatial*, *attention*, *mentalizing*, and *relevance* showed relatively high correlations with the right C4, suggesting that the left C4 could be more involved in memory and language, whereas the right C4 could be more involved in spatial attention and social functions.

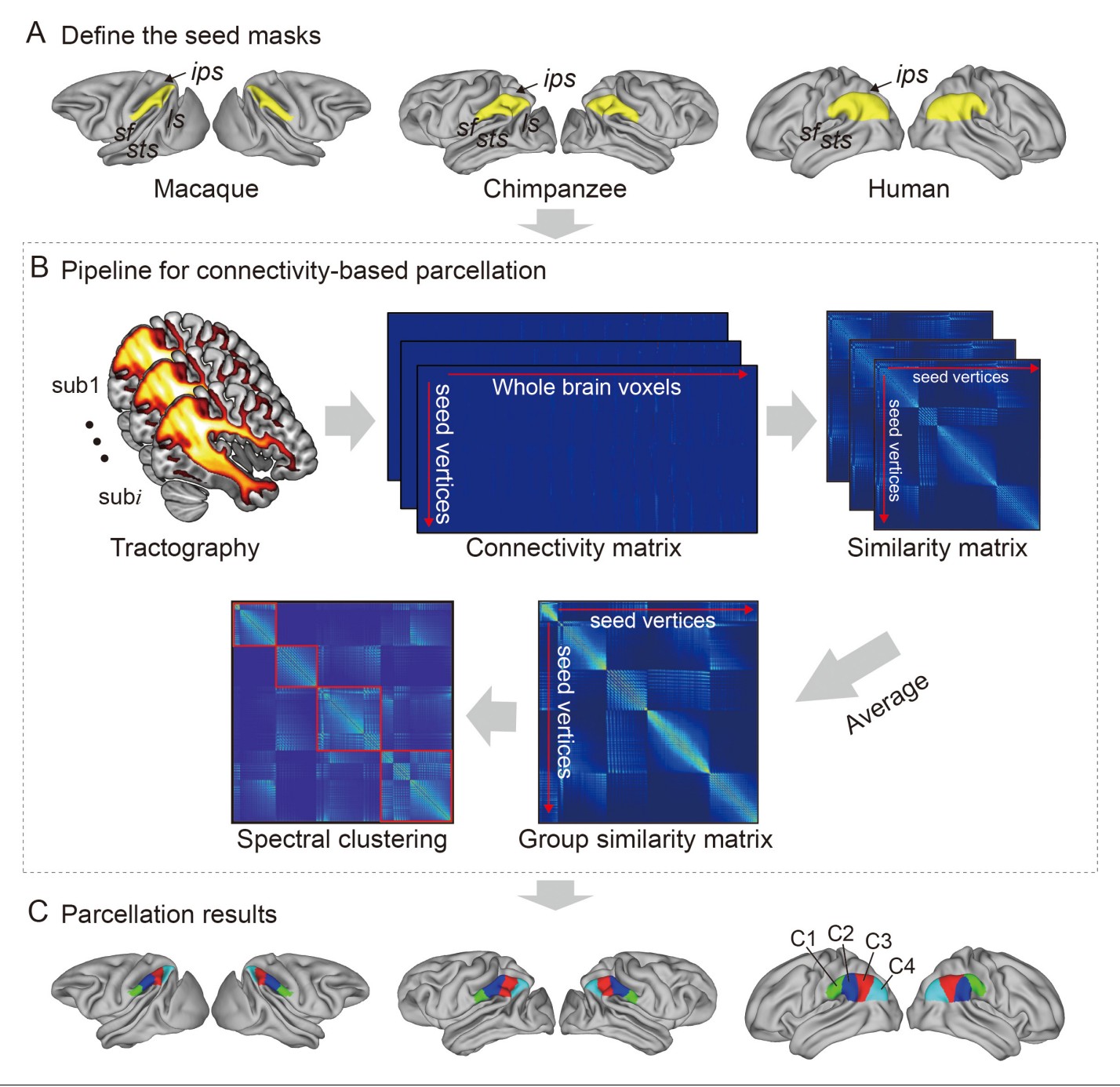

**Figure 1.** Framework of the connectivity-based brain parcellation for macaques, chimpanzees, and humans. (**A**) Defining the seed masks of the inferior parietal lobule (IPL) in surface space according to the gyri and sulci. (**B**) Connectivity-based parcellation using anatomical connectivity. Probabilistic tractography was applied by sampling 5000 streamlines at each vertex within the seed mask. Whole-brain connectivity profiles were used to generate a connectivity matrix with each row representing the connectivity profile of each seed vertex. Next, a correlation matrix was calculated as a measure of similarity between the seed vertices. Then, a group similarity matrix was calculated by averaging the correlation matrix across subjects and spectral clustering was applied to it. (**C**) Parcellation results of the IPL across species. The entire framework was applied independently for each hemisphere and each species.

The online version of this article includes the following figure supplement(s) for figure 1:

**Figure supplement 1.** Two- to five-cluster parcellation of the IPL.

**Figure supplement 2.** Functional decoding of the human left (**A**) and right (**B**) inferior parietal lobule (IPL) subregions.

### Allometric scaling and structural asymmetry of IPL subregions

When examining the relationship of the volume of each of the IPL subregions scaled against the total gray matter volume, the scaling of all the IPL subregions showed positive allometry (all slopes > 1) (*Figure 2A*). A statistical analysis revealed no significant differences between the slopes of each pair of the bilateral IPL subregions. The asymmetry indices (AIs) for the IPL subregions were calculated and are shown in *Figure 2B*. The macaques showed no significant asymmetry after Bonferroni correction for any of the subregions. The chimpanzees and humans both displayed leftward asymmetry in the rostral IPL (the C1 and C2, all p<0.001) and rightward asymmetry in the caudal IPL (the C3 and C4, all p<0.001).

### Connectional asymmetries of IPL subregions

To investigate the connectional asymmetries of the IPL subregions, we first calculated the connectivity profiles of the left and right subregions in macaques, chimpanzees, and humans using probabilistic tracking (*Figure 3—figure supplement 1*). Visualization of the connectivity patterns of the IPL did not show significant interhemispheric asymmetry in macaque monkeys or chimpanzees but did in humans, especially in connections with the inferior frontal gyrus (IFG) and lateral temporal cortex. A vertex-wise analysis was then performed to examine the connectional asymmetry of each subregion for each species by calculating the AIs between its connectivity profiles for the two hemispheres (*Figure 3*). Additionally, ROI- and tract-wise analyses were used to examine the asymmetry of the cortical regions and subcortical white matter pathways connected to the subregions, respectively (*Figure 4*; connectivity values shown in *Figure 4—figure supplements 1* and *2*). No significant asymmetries were found in macaques in any of the statistical analyses after correction for multiple comparisons.

In chimpanzees, the C1 showed significant leftward asymmetry mainly in connections with the anterior middle frontal gyrus (MFG), anterior IFG, planum temporale, and insula. The C2 showed significant leftward asymmetric connections with the insula and rightward asymmetric connections with

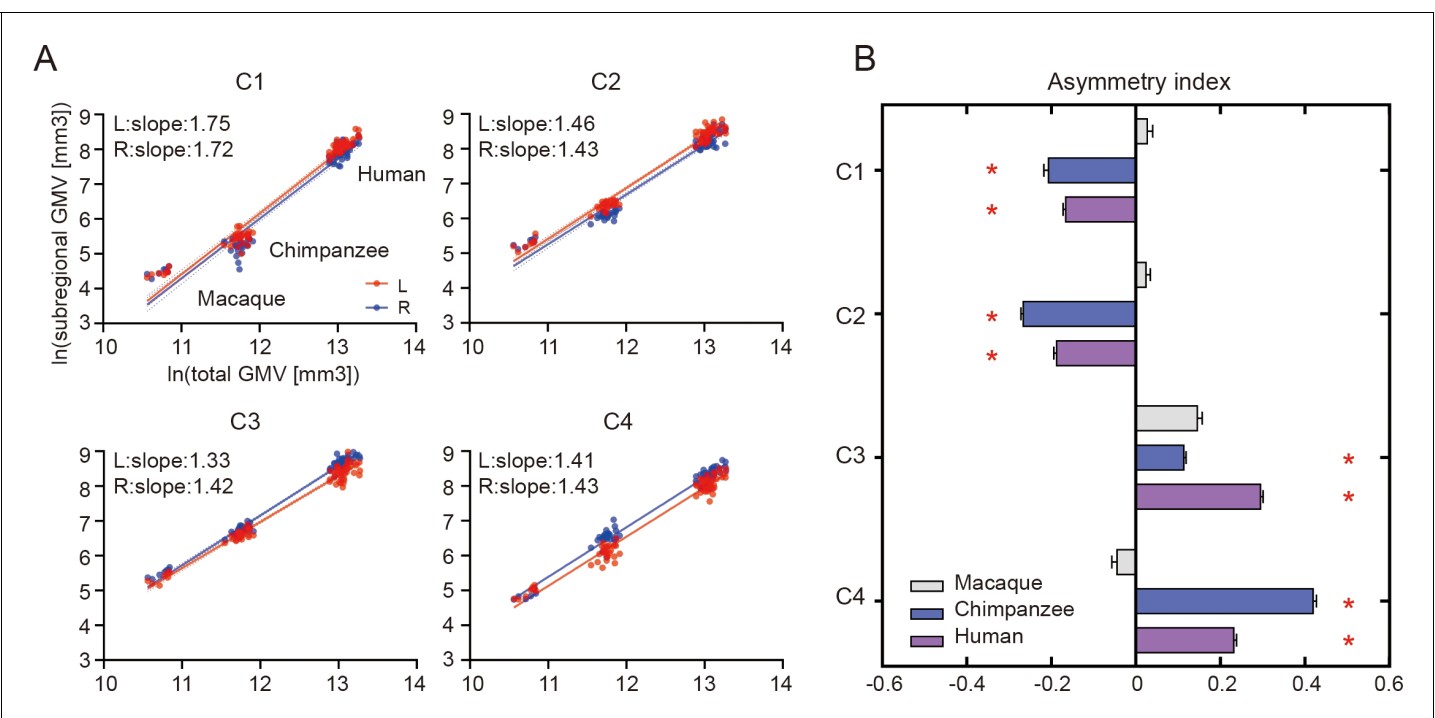

**Figure 2.** Structural allometric scaling and asymmetries of the inferior parietal lobule (IPL) subregions across species. (**A**) Volumes of the IPL subregions plotted against total cortical gray matter volume (GMV). Solid lines represent the best fit using mean macaque, chimpanzee, and human data points; dotted lines represent 95% confidence intervals. (**B**) Volumetric asymmetries of the IPL subregions. Negative asymmetry index indicates leftward asymmetry and positive index indicates rightward asymmetry. *Significance at the Bonferroni-corrected level of p<0.05. The error bars indicate the standard error of the mean.

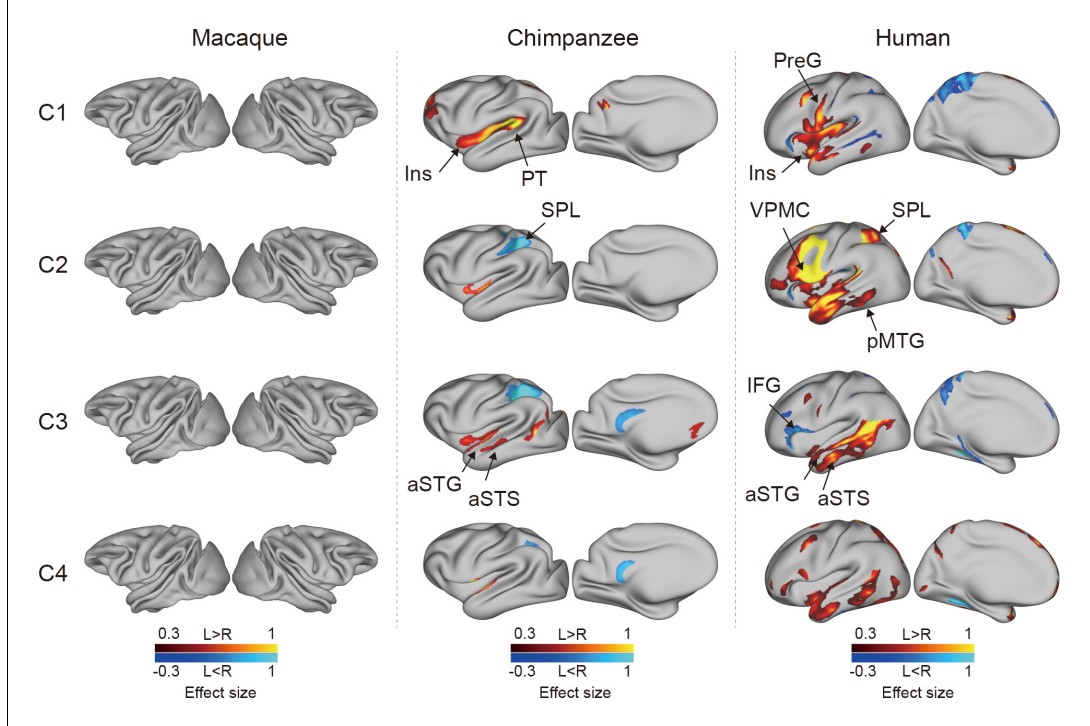

**Figure 3.** Connectional asymmetries of the IPL subdivisions in the vertex-wise analyses across species. Effect size (Cohen's d) related to asymmetric connections of IPL subdivisions displayed on the left hemisphere of a species-specific standard brain (leftward asymmetry: yellow, rightward asymmetry: blue) for each species for areas showing significance at the p<0.05 level corrected for multiple comparisons using false discovery rate correction. PreG, precentral gyrus; SPL, superior parietal lobule; aSTG, anterior superior temporal gyrus; aSTS, anterior superior temporal sulci; PT, planum temporale; VPMC, ventral premotor cortex; pMTG, posterior middle temporal gyrus; IFG, inferior frontal gyrus; Ins, insula.

The online version of this article includes the following figure supplement(s) for figure 3:

**Figure supplement 1.** Connectivity profiles of IPL subdivisions across species.

the superior parietal lobule (SPL) and superior longitudinal fasciculus 2 (SLF2). The C3 showed significant leftward asymmetric connections with the anterior superior temporal gyrus (STG), anterior superior temporal sulcus (aSTS), and occipitotemporal area and rightward asymmetric connections with the SPL and posterior cingulate gyrus (PCC). The C4 showed significant leftward asymmetric connections with the anterior STG (aSTG) and rightward asymmetry with the SPL and PCC.

In humans, the C1 showed significant leftward asymmetric connections with the ventral premotor and motor cortices and insula, which was consistent with regional leftward asymmetric connections with the precentral gyrus (PreG) and insula. The C1 also showed significant leftward asymmetric connections with the posterior MFG, aSTG, and posterior middle temporal gyrus (MTG) and rightward asymmetric connections with the orbital part of the IFG, posterior STS, and dorsal precuneus. The C2 showed significant leftward asymmetric connections with the posterior MFG, ventral premotor and motor cortices, SPL, anterior temporal lobe, and posterior MTG, which was consistent with regional leftward asymmetric connections with the IFG, PreG, postcentral gyrus (PostG), SPL, and STG and was supported by leftward asymmetric subcortical connections with the SLF2, SLF3, and arcuate fasciculus (AF). The C2 also showed rightward asymmetric connections with the orbital part of the IFG and posterior cingulate sulcus. The C3 showed significant leftward asymmetry mainly in the connections with the anterior IFG, SPL, and almost all the lateral temporal cortex, which was consistent with regional leftward asymmetric connections with the MTG and inferior temporal gyrus (MTG/ITG). The C3 also showed rightward asymmetric connections with the IFG, which was supported by leftward asymmetric subcortical connections with the SLF3. The C4 showed significant leftward asymmetry mainly in the connections with the IFG and anterior and posterior temporal cortex. The C4 also showed significant regional leftward asymmetric connections with the PreG, PostG, and SPL in the ROI-wise analysis.

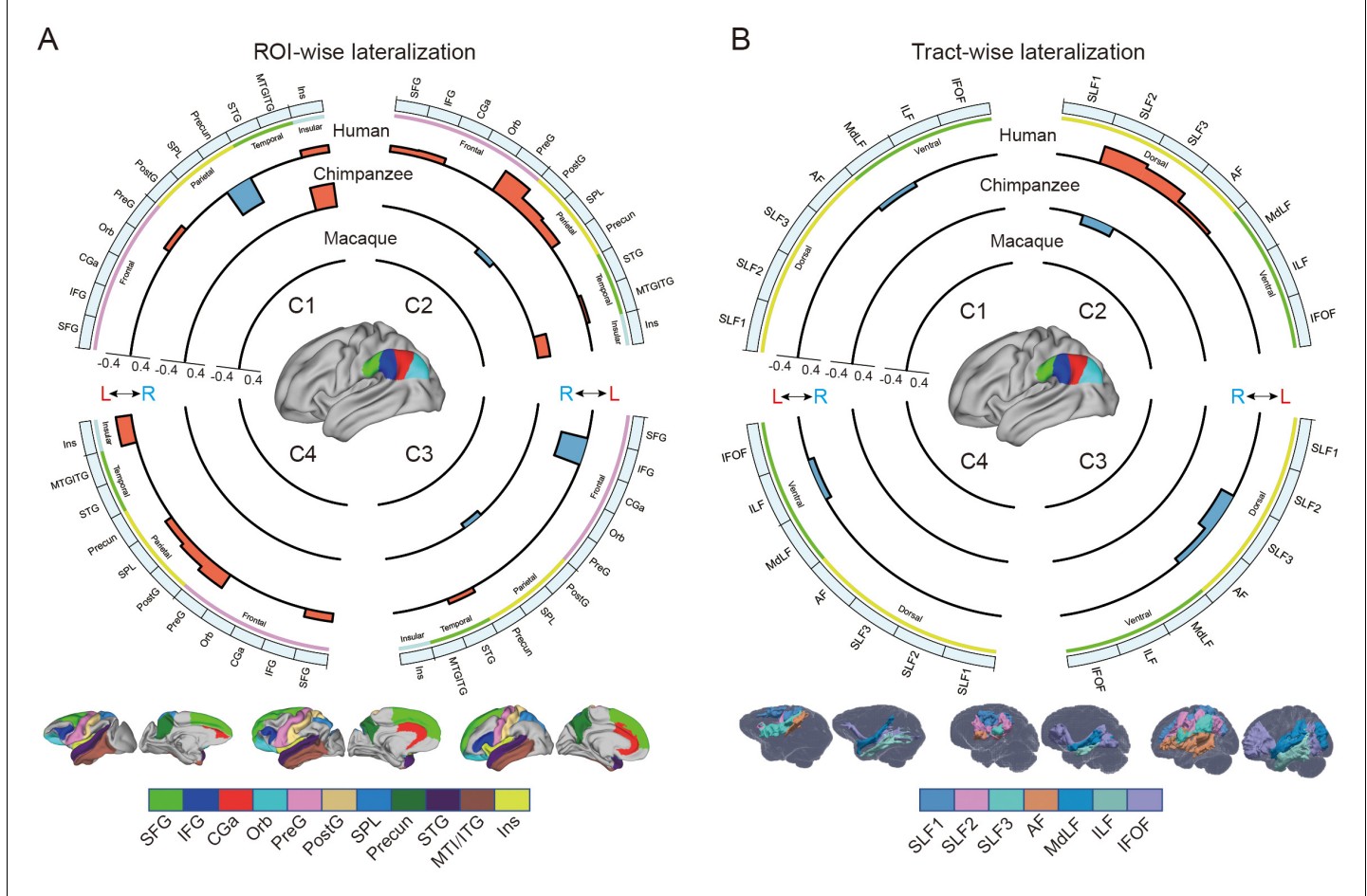

**Figure 4.** Regional connectional asymmetries of IPL subdivisions. (**A**) Connectional asymmetries of IPL subdivisions in the region of interest (ROI)-wise analyses across species. Connectional asymmetry was calculated for the connections between each IPL subregion and 11 ROIs. (**B**) Connectional asymmetries of the IPL subdivisions in the tract-wise analysis across species. Connectional asymmetry was calculated for the connections between each IPL subregion and the seven tracts. For all plots, the four quadrants of each circle correspond to the four IPL subregions. The outermost circles represent ROIs or tracts. The three inner circles from inside to outside represent macaques, chimpanzees, and humans, respectively. For all plots, only the connectivity showing a significance at a Bonferroni-corrected level of p<0.05 are displayed. SFG, superior frontal gyrus; IFG, inferior frontal gyrus; CGa, anterior cingulate gyrus; Orb, orbitofrontal cortex; PreG, precentral gyrus; PostG, postcentral gyrus; SPL, superior parietal lobule; STG, superior temporal gyrus; MTG/ITG, middle temporal gyrus and inferior temporal gyrus; Ins, insula; SLF1, SLF2, SLF3, the three branches of the superior longitudinal fasciculus; AF, arcuate fasciculus; MdLF, middle longitudinal fasciculus; ILF, inferior longitudinal fasciculus; IFOF, inferior fronto-occipital fasciculus.

The online version of this article includes the following figure supplement(s) for figure 4:

**Figure supplement 1.** Bar graphs of the average connectivity values between the inferior parietal lobule (IPL) subregions and 11 cortical regions for each species.

**Figure supplement 2.** Bar graphs of the average connectivity values between the inferior parietal lobule (IPL) subregions and 11 subcortical tracts for each species.

## Discussion

In the present study, we investigated asymmetries of the IPL in the structure and connectivity of rhesus macaques, chimpanzees, and humans. In the structural analysis, the IPL and its subregions exhibited a similar pattern of positive allometric scaling between hemispheres. In addition, the chimpanzees and humans shared similar asymmetric patterns in the IPL subregions, i.e., left asymmetry in the anterior part and right asymmetry in the posterior part, whereas macaques did not display asymmetry. In the connectivity analysis, the chimpanzees showed some connectional asymmetric regions including the SPL, insula, planum temporale, aSTG, and aSTS. The humans showed widespread connectional asymmetric regions including the primary motor and premotor cortices, SPL,

insula, and the entire lateral temporal lobe. These regions are associated with language, tool use, and handedness, suggesting a potential relationship between the connectional asymmetry and the functional hemispheric specialization of the human brain.

## Positive allometric scaling and structural asymmetry of IPL subregions

Brain allometry describes the quantitative scaling relationship between changes in the size of one structure relative to another structure, often the whole brain or cerebral cortex (*Mars et al., 2017*; *Smaers et al., 2017*). Previous allometric studies suggested that the association cortex (prefrontal, temporal, and parietal regions) scales with positive allometry (i.e., increases in size disproportionally, or more rapidly) across primates (*Passingham and Smaers, 2014*; *Mars et al., 2017*). Utilizing parcellation-based delineations, a recent study provided evidence that human brains have a greater proportion of prefrontal cortex gray matter volume than other primates (*Donahue et al., 2018*), and other studies demonstrate that human prefrontal expansion is greater than would be expected from allometric scaling in nonhuman primates (*Passingham and Smaers, 2014*; *Smaers et al., 2017*), although some conflicting analyses remain (*Gabi et al., 2016*). In the present study, we used macro-anatomical boundaries to identify the boundaries of the IPL and a connectivity-based parcellation approach to subdivide the IPL, which helped to reveal its internal organization. We found that the bilateral IPL subregions exhibited consistent, positive allometric scaling, which suggests that allometric scaling of the internal organization of the IPL was similar and was also consistent between homotopic regions during the evolution of the IPL in anthropoid primates. With only three species in the sample, our dataset does not allow us to use phylogenetic comparative statistical methods or determine whether human IPL subregions fall significantly above allometric expectations from nonhuman primates; future research that incorporates a broad phylogenetic sample of diverse primate brains would be necessary.

We found that chimpanzees and humans showed a similar dichotomous asymmetric pattern in their IPL subregions, that is leftward asymmetry in the anterior portion (the C1 and C2) and rightward asymmetry in the posterior portion (the C3 and C4), but macaques did not show any asymmetry. The result in humans is consistent with a recent study using data from a large consortium showing leftward asymmetry in the supramarginal gyrus and rightward asymmetry in the angular gyrus in terms of surface area (*Kong et al., 2018*). The divergent volumetric asymmetries suggest functional heterogeneities of the IPL and emphasize the importance of analyzing subregions within the IPL. The shared asymmetric pattern also suggests that divergences in the internal organization of the IPL evolved prior to the common ancestor of chimpanzees and humans and after the common ancestor of the three species.

## Connectional asymmetries underlying human language and complex tool use

Recent neuroimaging studies have highlighted specific brain regions and pathways that may be necessary for tool use (*Lewis, 2006*; *Stout and Chaminade, 2012*). We found that humans showed leftward asymmetric connectivity between the IPL (the C2) and the primary motor cortex, ventral premotor cortex, SPL, and posterior MTG, all of which were activated in tasks related to tool use and might constitute a cortical network underlying complex tool use (*Lewis, 2006*). In addition, portions of this network appeared to represent part of a system that is tightly linked with language systems. The interaction between the tool use system and the language system, though with a clear left hemisphere bias, is responsible for representing semantic knowledge about familiar tools and their uses and for acquiring the skills necessary to perform these actions (*Johnson-Frey, 2004*; *Lewis, 2006*; *Stout and Chaminade, 2012*; *Mars et al., 2017*). Several theories suggest that the evolutionary path leading to language and tool use in humans may be built upon modifications of circuits that subserve gestures and imitation (*Lewis, 2006*). Macaques are thought to emulate the goals and intentions of others, whereas chimpanzees can also imitate certain specific actions, but humans have an even stronger bias for high-fidelity copying of precise sequences of actions, which has been called 'overimitation' (*Hecht et al., 2013*). Our findings provide a potential explanation for these phenomena in that the macaques showed no asymmetric network connections, the chimpanzees showed a few asymmetric connections, but the humans showed a large number of asymmetric

connections. These species differences in leftward asymmetric connections involving language and tool use may reflect human specializations for language and complex tool use.

Unlike the humans, who showed considerable leftward asymmetry connectivity between the IPL and the lateral temporal cortex, the chimpanzees showed few leftward asymmetric connections between the IPL and the temporal cortex, including the planum temporale, aSTG, and aSTS, while macaques showed symmetric connections between the IPL and temporal cortex. The planum temporale is considered to include part of Wernicke's area homolog (*Spocter et al., 2010*) and displays leftward anatomical asymmetry in humans and great apes (*Gannon et al., 1998*; *Hopkins et al., 1998*). Recent work suggested a left-hemispheric size predominance of the planum temporale in olive baboons, a nonhominid primate species (*Marie et al., 2018*). We speculate that this planum temporale asymmetry may not be the only prominent characteristic related to language lateralization. The patterns from symmetry in macaques to asymmetry in humans and chimpanzees in the present study provide a possible new evidence that neural connectivity asymmetry may underlie the roots of language specialization, with the initial emergence of hemispheric specializations in apes which are elaborated even further in human brain evolution. In addition, identifying increased asymmetric connections between the IPL and planum temporale in human brains compared to chimpanzees and macaques reinforces the evidence that the evolutionary origin of human language capacities is related to further left-hemispheric specialization of neural substrates for auditory processing that are shared with other primates (*Balezeau et al., 2020*). Since the aSTG and aSTS have been implicated in semantic and phonologic processing in humans (*Vigneau et al., 2006*), the leftward asymmetric connections of the IPL with the aSTG and aSTS may be relevant to the evolution of human language processing. Our results suggest an evolutionary trajectory for the connectivity of the IPL with the temporal cortex; that is, the connectivity started as symmetric in macaque monkeys, began to develop asymmetrically in chimpanzees, and finally achieved the greatest degree of asymmetry and is refinement in humans. This sequence may support the emergence of language and language-related functions.

## Species-specific differences in asymmetric connectivity in chimpanzees and humans

Species-specific differences in asymmetric connectivity between the IPL and SPL were found in chimpanzees and humans, with leftward asymmetry in the former and rightward asymmetry in the latter, whereas no asymmetry of this connectivity was found in macaques. These species differences in hemispheric asymmetry may reflect evolutionary changes responsible for adaptations or the production of new abilities in the human brain. Structurally, in chimpanzees, right anatomical asymmetry in the white matter below the SPL (*Hopkins et al., 2010*) may increase the right connectivity between the IPL and SPL compared with the left side. In humans, the leftward volumetric asymmetry in the SPL (*Goldberg et al., 2013*), together with leftward volumetric asymmetry in the IPL (the C2), may support the leftward asymmetric connectivity. Functionally, interaction between the IPL and SPL is crucial for tool use, which is dominant in the left hemisphere, and visuospatial function, which is dominant in the right hemisphere (*Lewis, 2006*; *Thiebaut de Schotten et al., 2011a*; *Catani et al., 2017*). As for tool use, in contrast to the relatively simple tools used by chimpanzees and other species, humans can create complex artifacts through a sequence of actions that may incorporate multiple parts, reflecting a deep understanding of the kinematics of our bodies, the mechanical properties of surrounding objects, and the unique demands of the external environments in which we live (*Povinelli et al., 2000*; *Johnson-Frey, 2004*). In addition, complex tool use requires the SPL to code the location of the limbs relative to other body parts during planning and executing tool-use movements or hand gestures (*Wolpert et al., 1998*; *Johnson-Frey et al., 2005*; *Lewis, 2006*). Leftward asymmetric connectivity between the IPL and SPL may have provided a connectional substrate for complex tool use during human evolution. As for visuospatial functions, the rightward asymmetric connectivity between the IPL and SPL in chimpanzees may indicate that visuospatial functions are dominant in the right hemisphere and had already been lateralized to the right hemisphere from the common ancestor with macaques. During evolution, these lateralized functions may be retained in the human brain. Meanwhile, the lateralized directional reversal of this connectivity from the right to the left hemisphere may reflect evolutionary adaptations for the emergence of new abilities, such as sophisticated and complex tool making and use.

## Human unique asymmetric connectivity of IPL subregions

Unlike the chimpanzees and macaques, humans showed leftward asymmetry in the connection between the rostral IPL (the C1 and C2) and the primary motor cortex, which is consistent with a larger neuropil volume in the left primary motor cortex than in the right side (*Amunts et al., 1996*). Meanwhile, the leftward asymmetric volume of the anterior IPL and the primary motor cortex may also increase the neural connectivity between these two regions in the left hemisphere compared with the right side. Such a leftward connection is thought to be related to handedness and hand manual skills (*Amunts et al., 1996*; *Amunts et al., 1997*). In contrast to humans, chimpanzees and macaques did not show any asymmetric connectivity between the IPL and the primary motor cortex. A more recent study reported that, in olive baboons, contralateral hemispheric sulcus depth asymmetry of the central sulcus related to the motor hand area is correlated with the direction and degree of hand preference, as measured by a bimanual coordinated tube task, but only about 41% of them were classified as right handed and 33% were classified as left handed (*Margiotoudi et al., 2019*). Although previous studies have shown that chimpanzees exhibit population-level handedness in the use of tools and a corresponding asymmetry in the primary motor cortex, inferior frontal cortex, and parietal operculum (*Gilissen and Hopkins, 2013*; *Hopkins et al., 2017*), they do not show handedness as a more universal trait or exhibit manual dexterity to the same extent as humans. One possible explanation is that humans developed the asymmetric connectivity that became the structural basis for specific behaviors of handedness and hand skills during evolution.

An unexpected finding was that in humans the IPL, particularly the C3, showed rightward asymmetric connectivity with the IFG. Since the IPL and the IFG are interconnected through the SLF3, which is strongly rightward asymmetric (*Thiebaut de Schotten et al., 2011a*), it may also increase the connection between the IPL and IFG in the right hemisphere. Functionally, the left IFG is involved in various aspects of language functions, including speech production and semantic, syntactic, and phonological processing (*Wang et al., 2020*), whereas the right IFG is associated with various cognitive functions, including attention, motor inhibition, and social cognitive processes (*Hartwigsen et al., 2019*). Our result of rightward asymmetry in this connectivity seems to be associated with attention and social function, but not language, although language dominance in the left hemisphere is considered to be a common characteristic in humans.

The widespread asymmetric connections of the IPL in humans compared with the other two primates is in keeping with the interhemispheric independence hypothesis, in which, during evolution, brain size expansion led to hemispheric specialization due to time delays in neuron signaling over increasing distances, resulting in decreased interhemispheric connectivity and increased intra-hemispheric connectivity (*Ringo et al., 1994*; *Phillips et al., 2015*). While having more cortical neurons (local characteristics) in one hemisphere than the other seems to be a necessary condition for asymmetries of complex and flexible behaviors, it is not a full condition for such behaviors. Given that a function or behavior in an area is determined by its connectivity or networks in which it is involved (*Passingham et al., 2002*), the widespread lateralized connections may provide the human brain with the increased computational capacity necessary for processing language and complex tool use and may play a facilitating role in human cognitive and behavioral specialization.

## Methodological considerations

The three levels of analyses, i.e., the vertex-wise, ROI-wise, and tract-wise analyses, were performed to provide a full description of the connectivity asymmetry. However, it should be noted that some analyses produced results that were not completely consistent with each other. In humans, the connectivity of the IPL with SLF3 and AF was left-lateralized in the C2, while right-lateralized in the C3. The previous studies assessed the asymmetry of the SLF3 and AF with local characteristics such as cortical volume, voxel count, and fractional anisotropy and their average across all the voxels in the tracts (*Thiebaut de Schotten et al., 2011b*; *Thiebaut de Schotten et al., 2011a*; *Kamali et al., 2014*), the SLF3 and AF were usually found to have a single pattern, e.g., leftward asymmetry, rightward asymmetry, or symmetry. However, our results seem to indicate two different asymmetric patterns for the SLF3 and AF, both of which connect the IPL subregion C2 and C3 to the IFG (*Thiebaut de Schotten et al., 2011a*; *Hecht et al., 2015*; *Barbeau et al., 2020*). Furthermore, these connectivity asymmetries matched well with the ROI-wise and tract-wise analyses. The leftward connectivity asymmetry of the human C2 with the SLF3 and AF using the tract-wise approach

corresponds to that of the human C2 with the IFG and PreG using the ROI-wise approach. The rightward connectivity asymmetry of the human C3 with the SLF3 and AF using the tract-wise approach corresponds to that of the human C3 with the IFG using the ROI-wise approach. The SLF3, located at the ventrolateral SLF, connects to the IPL, especially the anterior part, and from there predominantly to the ventral premotor and prefrontal areas (*Thiebaut de Schotten et al., 2011a*; *Kamali et al., 2014*; *Barbeau et al., 2020*). The C2 and C3 appear to separate the SLF3 into two finer components, one connecting the posterior IFG and anterior IPL with leftward asymmetry and other connecting the anterior IFG to the posterior IPL with rightward asymmetry. The two types of connectivity patterns are consistent with previous studies using invasive tract-tracing findings in macaque monkeys and resting-state functional connectivity results in humans to study frontal and parietal connectivity (*Petrides and Pandya, 2009*; *Margulies and Petrides, 2013*). Our results indicated that both cortical areas, such as the IFG, and subcortical tracts, such as the SLF3 and AF, have at least two distinct subcomponents.

The inconsistency was observed when significant ROI-wise connectivity asymmetry was found, but few or no significant tract-wise connectivity asymmetries were found. For example, the human C1 showed ROI-wise connectivity asymmetry with the PreG and insula but no significant tract-wise connectivity asymmetry. The IPL is connected to the PreG mainly through the SLF3, which in turn is connected to not only the PreG but also the IFG and MFG in the prefrontal cortex (*Thiebaut de Schotten et al., 2011a*; *Kamali et al., 2014*; *Hecht et al., 2015*; *Barbeau et al., 2020*). The connectivity between the C1 and the PreG may include only a portion of the SLF3; this may have diluted the laterality effect from the SLF3 because it may include other pathways that were not in our selected tracts and could, thus, have affected the observed lateralization. In other words, the traditionally defined major fiber tracts are not a single bundle but, instead, contain many subcomponents. Therefore, the patterns of lateralization might not yet have been fully explored in our study. A recent work also suggested that the SLF2, SLF3, and AF could be separated into several branches based on their projections into the prefrontal and/or temporal areas (*Barbeau et al., 2020*). This may be true for the other major fiber tracts, such as the ILF (*Latini et al., 2017*), uncinate fasciculus (*Hau et al., 2017*), and cingulum bundle (*Jones et al., 2013*). On the other hand, brain regions, such as the IFG were connected to many fiber tracts, including the SLFII, SLFIII, and AF. Hence, we did not find tract-wise connectivity asymmetry that corresponded to the ROI-wise connectivity asymmetry in the human C1. This was also the case for the chimpanzee and human C4. The creation of a finer tract atlas should be a priority for future work because this would help to map the tract-wise connectivity asymmetry at a higher resolution.

In conclusion, we identified similar topographical maps of the IPL to study structural and connectional asymmetries in macaques, chimpanzees, and humans. We found that the structural asymmetry of the IPL was independent of the allometric scaling of this region. The connectional analysis revealed that humans had the largest connectional asymmetries of IPL subregions compared to macaques and chimpanzees. The regions showing larger asymmetric connections with the human IPL were associated with language, complex tool use, and handedness, which provided potential anatomical substrates for functional and behavioral lateralization in humans. The opposite asymmetric connection between the IPL and SPL in chimpanzees and humans may reflect distinct species-specific modifications to cortical circuits during the course of ape and human evolution.

## Materials and methods

### Human data

Data from 40 right-handed healthy adults (age: 22–35, 18 males) were randomly selected from the S500 subjects release of the Human Connectome Project (HCP) database (*Van Essen et al., 2013*) (http://www.humanconnectome.org/study/hcp-young-adult/). T1-weighted (T1w) MPRAGE images (resolution: 0.7 mm isotropic, slices: 256; field of view: 224 × 320; flip angle: 8°) and diffusion-weighted images (DWI) (resolution: 1.25 mm isotropic; slices: 111; field of view: 210 × 180; flip angle: 78°; b-values: 1000, 2000, and 3000 s/mm$^2$) were collected on a 3 T Skyra scanner (Siemens, Erlangen, Germany) using a 32-channel head coil.

## Chimpanzee data

Data from 27 adult chimpanzees (*Pan troglodytes*, 14 males) were made available by the National Chimpanzee Brain Resource (http://www.chimpanzeebrain.org, supported by the NIH National Institute of Neurological Disorders and Stroke). Data, including T1w and DWI, were acquired at the Yerkes National Primate Research Center (YNPC) on a 3T MRI scanner under propofol anesthesia (10 mg/kg/h) using previously described procedures (*Chen et al., 2013*). All procedures were carried out in accordance with protocols approved by YNPRC and the Emory University Institutional Animal Care and Use Committee (approval no. YER-2001206).

DWI were acquired using a single-shot spin-echo echo-planar sequence for each of 60 diffusion directions (b = 1000 s/mm$^2$, repetition time 5900 ms; echo time 86 ms; 41 slices; 1.8 mm isotropic resolution). DWI with phase-encoding directions (left–right) of opposite polarity were acquired to correct for susceptibility distortion. For each repeat of a set of DWI, five b = 0 s/mm$^2$ images were also acquired with matching imaging parameters. T1w images were also acquired for each subject (218 slices, resolution: $0.7 \times 0.7 \times 1$ mm).

## Macaque data

Data from eight male adult rhesus macaque monkeys (*Macaca mulatta*) were obtained from TheVirtualBrain (*Shen et al., 2019*). All surgical and experimental procedures were approved by the Animal Use Subcommittee of the University of Western Ontario Council on Animal Care (AUP no. 2008–125) and followed the Canadian Council of Animal Care guidelines. Surgical preparation and anesthesia as well as imaging acquisition protocols have been previously described (*Shen et al., 2019*). Images were acquired using a 7 T Siemens MAGNETOM head scanner. Two diffusion-weighted scans were acquired for each animal, with each scan having opposite phase encoding in the superior–inferior direction at 1 mm isotropic resolution, allowing for correction of susceptibility-related distortion. For five animals, the data were acquired with 2D EPI diffusion, while for the remaining three animals, a multiband EPI diffusion sequence was used. In all cases, data were acquired with b = 1000 s/mm$^2$, 64 directions, 24 slices. Finally, a 3D T1w image was also collected for each animal (128 slices, resolution: 0.5 mm isotropic).

## Image preprocessing

The human T1w structural data had been preprocessed following the HCP's minimal preprocessing pipeline (*Glasser et al., 2013*), while the chimpanzee and monkey T1w structural data had been preprocessed following the HCP's nonhuman preprocessing pipelines described in previous studies (*Glasser et al., 2013*; *Donahue et al., 2018*). Briefly, the processing pipeline included imaging alignment to standard volume space using FSL, automatic anatomical surface reconstruction using Free-Surfer, and registration to a group average surface template space using the Multimodal Surface Matching (MSM) algorithm (*Robinson et al., 2014*). Human volume data were registered to Montreal Neurological Institute (MNI) standard space and surface data were transformed into surface template space (fs_LR). Chimpanzee volume and surface data were registered to the Yerkes29 chimpanzee template (*Donahue et al., 2018*). Macaque volume and surface data were registered to the Yerkes19 macaque template (*Donahue et al., 2018*).

Preprocessing of the diffusion-weighted images was performed in a similar way in the human, chimpanzee, and macaque datasets using FSL. FSL's DTIFIT was used to fit a diffusion tensor model for each of the three datasets. Following preprocessing, voxel-wise estimates of the fiber orientation distribution were calculated using Bedpostx, allowing for three fiber orientations for the human dataset and two fiber orientations for the chimpanzee and macaque datasets due to the b-value in the diffusion data.

## Definition of the IPL

The IPL, located at the lateral surface of the ventral posterior parietal lobe, is surrounded by several sulci including the Sylvian fissure, superior temporal sulcus (STS), and intraparietal sulcus (IPS) (*von Economo and Koskinas, 1925*; *Von Bonin and Bailey, 1947*; *Bailey et al., 1950*; *Pandya and Seltzer, 1982*). In the absence of detailed homologous definitions, it is necessary to use cytoarchitectonic delineations and macroscopic boundaries, such as gyri and sulci, that can be reliably identified in all species as the boundaries of the IPL. The ROI of the IPL was manually drawn on the

standard surface template using Connectome Workbench (*Glasser et al., 2013*). In the present study, we restricted the ROI to the lateral surface of the IPL and excluded the cortex buried in the sulci, especially the lateral bank of the IPS and the upper bank of the Sylvian fissure. Rostrally, the IPL borders the vertical line between the Sylvian fissure and the rostral lip of the IPS. Dorsally, the IPL borders the lateral bank of the IPS. Ventrally, the anterior ventral IPL borders the upper bank of the Sylvian fissure. The border of the posterior and ventral IPL is formed by the extension of the Sylvian fissure to the top end of the STS in chimpanzees and macaques but by the extension of the Sylvian fissure to the posterior end of the IPS in humans.

## Connectivity-based parcellation

We used a data-driven connectivity-based parcellation framework modified from *Fan et al., 2016* (*Figure 1*). All steps in the framework were processed on surface data because the surface-based method has advantages, such as cortical areal localization (*Coalson et al., 2018*), over the traditional approach and because the use of surface meshes is a straightforward way to improve existing tractography processing pipelines, such as the precise locations of streamline seeding and termination (*St-Onge et al., 2018*). The surface ROI was first registered to native surface using MSM (*Robinson et al., 2014*). The probabilistic tractography was performed on the native mesh representing the gray/white matter interface using Probtrackx. The pial surfaces were used as stop masks to prevent streamlines from crossing sulci. Five thousand streamlines were seeded from each of the white matter surface vertices in the seed region to estimate its whole-brain connectivity profile and were downsampled to 5 mm isotropic voxels to construct the native connectivity M-by-N, a matrix between all the IPL vertices (M) and the brain voxels (N). Based on the native connectivity matrix, a symmetric cross-correlation M-by-M matrix was calculated to quantify the similarity between the connectivity profiles of each IPL vertex. A group cross-correlation matrix was calculated by averaging the cross-correlation matrix across subjects.

Data-driven spectral clustering was applied to the group cross-correlation matrix to define the anatomical boundaries of the IPL. Spectral clustering can capture clusters that have complicated shapes, making them suitable for parcellating the structure of complicated brain regions such as the IPL. In addition, the spectral clustering algorithm was successfully used to establish the Brainnetome Atlas (*Fan et al., 2016*). However, the number of clusters must be defined by the experimenter when using this method. In the current study, we explored from 2 to 12 parcellations.

## Volumetric analysis of the IPL

The cortical gray matter volumetric measurements were calculated using Freesurfer. Total cortical volumes were determined by the space between the white and pial surfaces in native space. Each subregion drawn on standard surface space was registered to native surface space using an existing mapping between the two meshes. The volume of the IPL and its subregions was determined by averaging all the vertices for each subject.

## Functional decoding of each subregion of the human IPL

Each subregion was first mapped to MNI volume space using a ribbon-constrained method in Connectome Workbench. To decode the functions of each subregion, we used the automated meta-analysis database, Neurosynth (*Yarkoni et al., 2011*), to identify the terms that were the most associated with each subregion. The top five non-anatomical terms with the highest correlation values were kept for all subregions and redundant terms, such as '*semantic*' and '*semantics*', were only considered once. For simplicity, we only showed the positive correlations found by decoding because negative correlations do not directly inform us about the functions of the subregions. The lateralization for each term was obtained by calculating the difference in the correlation values of the subregions between the left and right hemispheres.

## Mapping anatomical connectivity profiles

To map the whole-brain anatomical connectivity pattern for each cluster, we performed probabilistic tractography by drawing 5000 samples from each vertex in each cluster. The resulting tractograms were log-transformed, normalized by the maximum, and then projected onto surface space using the 'surf_proj' command in FSL to obtain tractograms in surface space. The surface tractograms

were smoothed using a 4 mm kernel for humans, 3 mm kernel for chimpanzees, and 2 mm kernel for macaques. We subsequently averaged the surface tractograms across subjects for the left and right hemispheres separately to obtain population tractograms, which were thresholded by a value of 0.5 for humans, 0.2 for chimpanzees, and 0.3 for macaques due to data quality. The resultant population tractograms represented approximately 20% of the non-zero vertexes in the non-thresholded population tractograms and were used for the vertex-wise and ROI-wise comparisons. The volumetric tractograms were used for the tract-wise comparison.

## Vertex-wise analysis

For each subregion, we restricted the analysis to the group mask defined by the combination of the left and mirrored right population tractograms described above. We here used the connectivity probabilistic value to quantify the connectivity between the IPL and each vertex of the rest of the brain. A higher value in the vertex means a higher likelihood of being connected to the IPL than other vertices.

## ROI-wise analysis

Although previous studies have devoted much effort to establishing homologous regions in primates, these are still limited to a few regions, particularly in chimpanzees. To make comparisons across species possible, here we used the common principle of macroscopic anatomical boundaries based on the gyri and sulci to define ROIs in the cerebral cortex. Specifically, the Desikan–Killiany–Tourville (DKT) atlas was used for humans (*Desikan et al., 2006*), a modified DKT atlas for the chimpanzees, and the Neuromaps atlas for the macaques (*Rohlfing et al., 2012*). Because the Neuromaps atlas is volumetric, we first mapped it to surface space for the subsequent calculations. A total of 11 cortical ROIs were chosen for each hemisphere: the superior frontal gyrus, IFG (a combination of the pars triangularis and pars opercularis in humans and chimpanzees), anterior cingulate gyrus (CGa, a combination of the rostral and caudal anterior-cingulate in humans and chimpanzees), orbitofrontal cortex (Orb), PreG, PostG, SPL, precuneus, STG, middle temporal gyrus and inferior temporal gyrus (MTG/ITG), and insula. The MTG/ITG was a combination of the MTG and ITG in humans and chimpanzees due to the absence of the MTG in macaques. The connectional value for each ROI was calculated by averaging all vertices in the ROI on the individual surface tractogram for each subregion.

## Tract-wise analysis

To investigate which subcortical fiber tracts are associated with lateralization of cortical areas connected to the IPL, we analyzed the lateralization of the subcortical white matter tracts connected to the IPL across species. A total of seven tracts were chosen: the three branches of the superior longitudinal fasciculus, AF, middle longitudinal fasciculus, inferior longitudinal fasciculus, inferior fronto-occipital fasciculus. The automated tractographic protocols for tracts for each species were from previous studies (*Bryant et al., 2020*), and these tracts were reconstructed using the Xtract tool (*Warrington et al., 2020*). The mean value for each tract was calculated by averaging all voxels in the tract in the individual volumetric tractogram for each subregion.

## Statistical analysis

To investigate the allometric relationship between the volume of each of the IPL subregions and the total gray matter volume using log-transformed data (*Donahue et al., 2018*), linear regression was performed by pooling the human, chimpanzee, and macaque data for each of the IPL subregions, separately. To test whether the scaling regression slopes differed significantly between the two hemispheres, we performed an ANCOVA for comparisons across the two regression slopes for each plot.

In all the analyses of the structural and connectional asymmetries (i.e., volumetric, vertex-wise, ROI-wise, and tract-wise), the AI was defined as the difference between values for the left and right hemispheres according to the formula $AI = 2 \times (R - L) / (R + L)$. For the vertex-wise analysis, a one-sample t test was performed at each vertex on the group mask for each species using PALM, with 5000 permutations with a sign-flip strategy (*Winkler et al., 2014*). The statistically significant level was set at false discovery rate corrected p<0.05. The effect sizes (Cohen's d) were displayed on the

average surface. For the volumetric, ROI-wise, and tract-wise analysis, a two-sided Wilcoxon signed-rank test was performed for each subregion. Bonferroni correction was then used for multiple comparisons for seeds, ROIs or tracts, and species, with statistical significance set at p<0.05.

## Acknowledgements

This work was partially supported by the National Natural Science Foundation of China (Grant Nos. 91432302, 82072099, and 31620103905), the Science Frontier Program of the Chinese Academy of Sciences (Grant No. QYZDJ-SSW-SMC019), Beijing Municipal Science and Technology Commission (Grant Nos. Z161100000216139 and Z171100000117002), the Guangdong Pearl River Talents Plan (2016ZT06S220), Key-Area Research and Development Program of Guangdong Province (2018B030333001), the Strategic Priority Research Program of Chinese Academy of Sciences (XDB32030200), the Youth Innovation Promotion Association, the Beijing Advanced Discipline Fund, and the National Science Foundation (SMA-1542848). The National Chimpanzee Brain Resource was supported by NIH – National Institute of Neurological Disorders and Stroke (NIH Grant No. NS092988). We thank Rhoda E and Edmund F Perozzi, PhDs, for English language and editing assistance.

## Additional information

### Funding

| Funder | Grant reference number | Author |
|---|---|---|
| National Natural Science Foundation of China | 91432302 | Tianzi Jiang |
| National Natural Science Foundation of China | 82072099 | Lingzhong Fan |
| National Natural Science Foundation of China | 31620103905 | Tianzi Jiang |
| Science Frontier Program of the Chinese Academy of Sciences | QYZDJ-SSW-SMC019 | Tianzi Jiang |
| Guangdong Pearl River Talents Plan | 2016ZT06S220 | Tianzi Jiang |
| Key-Area Research and Development Program of Guangdong Province | 2018B030333001 | Tianzi Jiang |
| Youth Innovation Promotion Association of the Chinese Academy of Sciences | | Lingzhong Fan |
| Beijing Advanced Discipline Fund | | Gaolang Gong Lingzhong Fan |
| National Science Foundation | SMA-1542848 | Chet Sherwood |
| Beijing Municipal Science and Technology Commission | Z161100000216139 | Tianzi Jiang |
| Beijing Municipal Science and Technology Commission | Z171100000117002 | Lingzhong Fan |
| Strategic Priority Research Program of Chinese Academy of Sciences | XDB32030200 | Lingzhong Fan Tianzi Jiang |

The funders had no role in study design, data collection and interpretation, or the decision to submit the work for publication.

## Author contributions
Luqi Cheng, Conceptualization, Methodology, Formal analysis, Investigation, Writing - original draft, Writing - review and editing; Yuanchao Zhang, Jiaojian Wang, Investigation, Writing - review and editing; Gang Li, Formal analysis; Chet Sherwood, Resources, Data curation, Writing - review and editing; Gaolang Gong, Writing - review and editing; Lingzhong Fan, Tianzi Jiang, Conceptualization, Supervision, Funding acquisition, Project administration, Writing - review and editing

## Author ORCIDs
Luqi Cheng (ID) https://orcid.org/0000-0002-4890-7424
Lingzhong Fan (ID) https://orcid.org/0000-0002-4813-621X
Tianzi Jiang (ID) https://orcid.org/0000-0001-9531-291X

## Ethics
Human subjects: The authors agreed to the Open Access Data Use Terms of the Human Connectome Project (Van Essen et al 2013). Informed consent from participating individuals was obtained by the Human Connectome Project investigators.
Animal experimentation: Chimpanzee data: The chimpanzee imaging data were acquired under protocols approved by the Yerkes National Primate Research Center (YNPRC) at Emory University Institutional Animal Care and Use Committee (Approval number YER2001206). Macaque data: All surgical and experimental procedures were approved by the Animal Use Subcommittee of the University of Western Ontario Council on Animal Care (AUP no. 2008-125) and followed the Canadian Council of Animal Care guidelines.

## Decision letter and Author response
Decision letter https://doi.org/10.7554/eLife.67600.sa1
Author response https://doi.org/10.7554/eLife.67600.sa2

# Additional files
## Supplementary files
• Transparent reporting form

## Data availability
The datasets analyzed during the current study are available at https://www.humanconnectome.org, https://www.chimpanzeebrain.org, https://openneuro.org/datasets/ds001875/versions/1.0.3, and https://www.neurosynth.org. All data generated or analysed during this study are included in the manuscript and supporting files. The resulting maps from this study are available at https://github.com/LuqiCheng/IPL_Connectional_Asymmetry (copy archived at https://archive.softwareheritage.org/swh:1:rev:2678b85cd75d73c42227035926d2cc388d2b122d, https://doi.org/10.5061/dryad.s4mw6m96m).

The following dataset was generated:

| Author(s) | Year | Dataset title | Dataset URL | Database and Identifier |
|---|---|---|---|---|
| Cheng L, Zhang Y, Li G, Wang J, Sherwood C, Gong G, Fan L, Jiang T | 2021 | Connectional Asymmetry of the Inferior Parietal Lobule Shapes Hemispheric Specialization in Humans, Chimpanzees, and Rhesus Macaques | https://doi.org/10.5061/dryad.s4mw6m96m | Dryad Digital Repository, 10.5061/dryad.s4mw6m96m |

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

## Appendix 1

### Functional decoding of human IPL subregions

To decode the functions of the human IPL subregions, we investigated the relationship of each sub-region with brain activation patterns obtained from the Neurosynth database (*Yarkoni et al., 2011*). The functional decoding of each subregion in each hemisphere is shown in *Figure 1—figure supplement 2*. To assess in which hemisphere a given functions was dominant, we calculated the differences in the correlation values for the left and right subregions (*Figure 1—figure supplement 2C*). The five most correlated terms for the left and right corresponding subregions were shown to be similar in some cases and dissimilar in others, suggesting functional lateralization in each subregion. These selected items exhibited a dichotomous pattern of asymmetry. Specifically, the left and right C1 both demonstrated relatively high correlations with sensory-related terms, such as *somatosensory*, *tactile*, *touch*, and *pain*. The term *tool* showed a prominently higher correlation with the left C1 compared to the right C1. The left C2 demonstrated relatively high correlations with terms including *grasping*, *monitoring*, *inhibition*, *semantics*, and *tool*, and the right C2 showed relatively high correlations with terms including *nogo*, *inhibition*, *preparatory*, *error*, and *detection*. The term *tool* and the language-related term *semantics* showed relatively high correlations with the left C2, whereas executive-related terms, such as *nogo* and *inhibition*, showed relatively high correlations with the right C2. The left C3 demonstrated relatively high correlations with terms including *retrieval*, *solving*, *recollection*, *judgments*, and *coherent*, and the right C3 showed relatively high correlations with terms including *beliefs*, *intentions*, *mentalizing*, *monitoring*, and *perspective*. The memory- and language-related terms, such as *retrieval*, *episodic*, *recollection*, *memories*, and *coherent*, showed relatively high correlations with the left C3, whereas executive-related terms, such as *nogo* and *inhibition*, and the social-related term, such as *beliefs*, were correlated with the right C3. The left C4 demonstrated relatively high correlations with terms including *episodic*, *autobiographical*, *retrieval*, *coherent*, and *memories*, and the right C4 showed relatively high correlations with terms including *spatial*, *attention*, *mentalizing*, *retrieval*, and *relevance*. The memory-related term *episodic* and the language-related term *coherent* showed relatively high correlations with the left C4, whereas the attention- and social-related terms, such as *spatial*, *attention*, *mentalizing*, and *relevance*, were correlated with the right C4.

### Connectivity profiles of IPL subregions

The whole-brain connectivity profiles of the left and right subregions were mapped in macaques, chimpanzees, and humans using probabilistic tracking (*Figure 3—figure supplement 1*). In macaques, all regions showed consistent connections with the ventral bank of the principal sulcus, the medial frontal cortex, superior parietal lobule (SPL), precuneus, superior temporal sulcus (STS), and insula. The C1, C2, and C3 were also connected with the premotor and sensorimotor cortex. The caudal subregions (the C3 and C4) were more connected with the superior temporal gyrus (STG). Visualization of the connectivity patterns did not show obvious interhemispheric asymmetry. In chimpanzees, all regions were connected with the middle frontal gyrus (MFG), inferior frontal gyrus (IFG), SPL, precuneus, planum temporale, and insula. The rostral subregions (the C1 and C2) were connected with the ventral part of the premotor and sensorimotor cortex, while caudal subregions (the C3 and C4) had strong connections with almost all the STG, the middle temporal gyrus (MTG), and the occipitotemporal areas. Visualization of the connectivity patterns did not show interhemispheric asymmetry. In humans, all regions showed connections with the IFG, the caudal part of the MFG, SPL, dorsal precuneus, and the lateral temporal cortex. The most rostral subregion C1 also showed connections with the primary motor and sensorimotor cortices and relatively few connections with the lateral temporal cortex compared with the other three subregions. The caudal subregion C4 also showed connections extending to the occipitotemporal areas. Visual inspection of the connectivity profiles of the IPL showed obvious interhemispheric asymmetries, especially in connections with the IFG and lateral temporal cortex.

