## [Decision Letter]

**Acceptance summary:**

This study examined asymetries in the inferior parietal lobe of two great ape species (humans and chimps) and a monkey species (macaque) based on analysis of the connectivity of the different subareas using diffusion MR and tract analysis. Four subareas are defined. The work demonstrates asymmetries in the connectivity from inferior parietal lobe in humans and chimps that are not present in the macaque. The work is important because it supports the evolution of asymmetrical pathways from the inferior parietal lobe that is already present in chimps that might be relevant to the evolution of language in man.

**Decision letter after peer review:**

Congratulations, we are pleased to inform you that your article, "Connectional Asymmetry of the Inferior Parietal Lobule Shapes Hemispheric Specialization in Humans and Nonhuman Primates", has been accepted for publication in *eLife*. Your article has been reviewed by 2 peer reviewers and the evaluation has been overseen by a Reviewing Editor and a Senior Editor.

The referees bring up some points of clarification and make suggestions about exposition you might want to consider before publication. I would particularly highlight referee #1 suggestion of a graphical abstract that I think is a helpful suggestion to aid readers.

*Reviewer #1:*

This paper from Luqi Cheng and colleagues explored the hemispheric specialization of the inferior parietal lobule in human and non-human primates, on structural and connectional level.

Using diffusion MRI scans of humans, chimpanzee and macaques, they applied an automatic connectivity based parcellation to obtain a comparable set of four subdivision of the IPL between the three species. They compared the allometric scaling of gray matter volume for each group, and found a common pattern of leftward and rightward lateralization in the four subdivisions of the IPL between humans and chimpanzees, while macaques were not significantly lateralized for any subregion. Then, comparing connectivity of the four subdivisions of IPL, they found widespread and mostly leftward lateralization of the IPL connectivity in humans, some to lesser extent in chimpanzee and no significant lateralization in macaques.

The most lateralized regions and tracts were found to be those involved in complex functions such as language and tool use in humans.

It is overall a well-designed and well conducted study. The parcellation method allows for a relevant comparison between species. While the segmentation of the IPL itself was done manually, the definition is clear and detailed. The three levels connectivity approach is interesting and allows a more comprehensive view than using a tract-wise only or ROI-wise only analysis. The results support recent works suggesting an emergence of hemispheric specialization in apes, and an important increase in lateralization in the human brain. It is a significant finding that provides a clearer view of the evolutionary story of the human brain.

There are however some points that could be improved:

1) The tract-wise approach shows unclear results which should be further discussed. There is a lack of concordance between ROIs connectivity and tract connectivity. Notably, in humans, C1 shows some ROIs based connectivity asymmetry, but no significant leftward asymmetry in the tracts connectivity. And in chimps, there is no significant tract-wise leftward asymmetry in any subregion of the IPL while it exists in the ROI-wise connectivity. It raises other intriguing questions such as: why are human SLF3 and arcuate left lateralized in C2 and right lateralized in C3? Or why do C3 show ROI-wise lateralization for but no tract-wise lateralization at all?

The results are what they are but an effort to link tract-wise connectivity to ROI-wise connectivity and to explain eventual discrepancies between them would benefit to the manuscript. Some of those somehow unexpected results could be biologically relevant, some might reveal limitations of a tract-based approach and make a case for a multiple approaches study like this one.

Furthermore, because it only shows a relative lateralization index, Figure 4 does not allow to tell whether a null value is the result of symmetry, or if it simply comes from the absence of connectivity.

2) The discussion focuses heavily on the chimpanzee-human distinction. I think the macaque difference with both apes species is a big story and should be equally highlighted as it is relevant to the question of the origin itself of the hemispheric lateralization. Which functional and behavioural features appeared when apes diverged from monkeys as their hemispheres began to specialize is of particular interest.

3) I think the manuscript is missing a figure carrying the central message of the study. To improve the impact, I recommend to add a graphical summary highlighting the main findings and putting them into the evolutionary context. It could be a schematic view of IPL and its projections in the 3 species, or/and a simplified phylogenic tree displaying the main evolutionary changes (for example, similar to figure 2. in Stout et al. [1]).

4) Figure 4: the connectional asymmetry alone is confusing as some connections or tracts may simply be absent or to weak to detect a significant asymmetry. The figure should somehow indicates the connection strength, or which connections are significant and which are not.

*Reviewer #2:*

Understanding how the key subdivisions of the brain, including parietal lobe, may have differentiated in human and nonhuman primates is of substantial importance. The authors focus on the asymmetry of the inferior parietal lobule, using a spectral clustering and connectivity analysis, and find a largely symmetrical pattern in macaques, with greater left and right asymmetry in chimpanzees and humans. Interestingly, the chimpanzee pattern is not identical or as robust as that in humans supporting the hypothesis of independent-hemispheric specialization in ancestors to apes and humans with substantial further hemispheric asymmetry developing more recently during human evolution.

The paper is well written, robustly analysed, compelling and the shared results will be extremely useful for the neuroscientific community. The only potential issue that I grappled with is the clustering of the IPL into 4 regions in each species using the similarity analyses. As they note current human, ape and macaque histological atlases for the IPL are difficult to compare and it is not clear which anatomical regions correspond across the species. The spectral similarity clustering approach is well suited for the paper, but the authors, again as they note, need to select an ideal cluster size, which could range from 1-12. However, in my view they selected a good cluster size value of 4 because with 5 the additional IPL cluster is in the wrong dorsal/ventral location in humans/chimpanzees, which would be very difficult to understand from an evolutionary perspective. So, the authors do a fair job dealing with my concern. Still it would be good for the shared results to ensure that they can be easily cross-referenced to atlases in each of the species, or the results taken forward by the authors or others in the future with other parcellation schemes.

---

## [Author Response]

Reviewer #1:[…] 1) The tract-wise approach shows unclear results which should be further discussed. There is a lack of concordance between ROIs connectivity and tract connectivity. Notably, in humans, C1 shows some ROIs based connectivity asymmetry, but no significant leftward asymmetry in the tracts connectivity. And in chimps, there is no significant tract-wise leftward asymmetry in any subregion of the IPL while it exists in the ROI-wise connectivity. It raises other intriguing questions such as: why are human SLF3 and arcuate left lateralized in C2 and right lateralized in C3? Or why do C3 show ROI-wise lateralization for but no tract-wise lateralization at all?The results are what they are but an effort to link tract-wise connectivity to ROI-wise connectivity and to explain eventual discrepancies between them would benefit to the manuscript. Some of those somehow unexpected results could be biologically relevant, some might reveal limitations of a tract-based approach and make a case for a multiple approaches study like this one.

Thank you for your comment and suggestion. The human C1 showed ROI-wise connectivity asymmetry with the precentral gyrus and insula but no significant asymmetry in the tract-wise connectivity. Some potential reasons are: First, the ROI-wise connectivity is not completely overlapped with the predefined tracts. For example, the precentral gyrus is mainly connected to C1 subregion with SLFII and SLFIII, but the SLFIII is mainly connected C1 with the precentral gyrus and the IFG (De Schotten et al., 2011; Kamali et al., 2014). Second, subcomponents are present in brain regions and major fiber tracts, a concept which is supported by recent work suggesting that the SLF2, SLF3, and AF could be separated into dorsal and ventral branches (Barbeau et al., 2020), as could the ILF (Jones et al., 2013), uncinate fasciculus (Hau et al., 2017), and cingulum bundle (Latini et al., 2017). Hence, the measured connectivity of the SLF3 in the present study may only include a portion of the actual connectivity. Although the SLF3 was shown to be asymmetric (De Schotten et al., 2011), its laterality effect may be diluted, leading to no asymmetry in the tract-wise analysis. This is also the case in chimpanzees and the human C4.

The human SLF3 and AF were found to be left-lateralized in the C2 and right-lateralized in the C3, which seems to be contradictory but reasonable. The two connectivity asymmetries matched well in the ROI-wise and tract-wise analyses. The leftward connectivity asymmetry between the human C2 and the SLF3 and AF using the tract-wise approach corresponds to that between the human C2 and the IFG and precentral gyrus using the ROI-wise approach. The rightward connectivity asymmetry between the human C3 and the SLF3 and AF using the tract-wise approach corresponds to that between the human C3 and the IFG using the ROI-wise approach. These two connectivity findings were consistent with previous studies using invasive tract-tracing in macaque monkeys and resting-state functional connectivity in humans to study the frontal and parietal connectivity (Petrides and Pandya, 2009; Margulies and Petrides, 2013). Our results also provided more information showing that the two connectivities have different asymmetric patterns.

We have added some relevant text in the Discussion section as follows.

“Methodological considerations

The three levels of analyses, i.e., the vertex-wise, ROI-wise, and tract-wise analyses, were performed to provide a full description of the connectivity asymmetry. […] The creation of a finer tract atlas should be a priority for future work because this would help to map the tract-wise connectivity asymmetry at a higher resolution.”

Furthermore, because it only shows a relative lateralization index, Figure 4 does not allow to tell whether a null value is the result of symmetry, or if it simply comes from the absence of connectivity.

Thanks for your question. To clarify this, we have added appendix figures indicating the connectivity values for each subregion for each species (Figure 4—figure supplements 1 and 2).

2) The discussion focuses heavily on the chimpanzee-human distinction. I think the macaque difference with both apes species is a big story and should be equally highlighted as it is relevant to the question of the origin itself of the hemispheric lateralization. Which functional and behavioural features appeared when apes diverged from monkeys as their hemispheres began to specialize is of particular interest.

Thank you for your wonderful suggestion, we have added some text to highlight the differences between macaques and humans and chimpanzees as follows.

In the second paragraph of the “Connectional asymmetries underlying human language and complex tool use” of the Discussion – “[…], while macaques showed symmetric connections between the IPL and temporal cortex. […] This sequence may support the emergence of language and language-related functions.”

In the first paragraph of the “Human unique asymmetric connectivity of IPL subregions” of the Discussion – “A more recent study reported that, in olive baboons, contralateral hemispheric sulcus depth asymmetry of the central sulcus related to the motor hand area is correlated with the direction and degree of hand preference, as measured by a bimanual coordinated tube task, but only about 41% of them were classified as right-handed and 33% were classified as left-handed (Margiotoudi et al., 2019).”

3) I think the manuscript is missing a figure carrying the central message of the study. To improve the impact, I recommend to add a graphical summary highlighting the main findings and putting them into the evolutionary context. It could be a schematic view of IPL and its projections in the 3 species, or/and a simplified phylogenic tree displaying the main evolutionary changes (for example, similar to figure 2. in Stout et al. [1]).

Thank you for your suggestion. We have added a graphical abstract summarizing the main results in the present study (See Author response image 1).

**Author response image 1. respfig1:** The graphical abstract summarizing the main results in the present study.

Reviewer #2:[…] The paper is well written, robustly analysed, compelling and the shared results will be extremely useful for the neuroscientific community. The only potential issue that I grappled with is the clustering of the IPL into 4 regions in each species using the similarity analyses. As they note current human, ape and macaque histological atlases for the IPL are difficult to compare and it is not clear which anatomical regions correspond across the species. The spectral similarity clustering approach is well suited for the paper, but the authors, again as they note, need to select an ideal cluster size, which could range from 1-12. However, in my view they selected a good cluster size value of 4 because with 5 the additional IPL cluster is in the wrong dorsal/ventral location in humans/chimpanzees, which would be very difficult to understand from an evolutionary perspective. So, the authors do a fair job dealing with my concern. Still it would be good for the shared results to ensure that they can be easily cross-referenced to atlases in each of the species, or the results taken forward by the authors or others in the future with other parcellation schemes.

Thank you for your positive comments. We are planning to share the results including the parcellation schemes in the present study. They will be available from the Brainnetome Atlas website (https://atlas.brainnetome.org/index.html) and github (https://github.com/LuqiCheng/IPL_Connectional_Asymmetry).

References:

Barbeau EB, Descoteaux M, Petrides M. Dissociating the white matter tracts connecting the temporo-parietal cortical region with frontal cortex using diffusion tractography. Scientific reports 2020; 10(1): 1-13.De Schotten MT, Dell’Acqua F, Forkel S, Simmons A, Vergani F, Murphy DG, Catani M. A lateralized brain network for visuo-spatial attention. Nature Precedings 2011: 1-.Hau J, Sarubbo S, Houde JC, Corsini F, Girard G, Deledalle C, Crivello F, Zago L, Mellet E, Jobard G. Revisiting the human uncinate fasciculus, its subcomponents and asymmetries with stem-based tractography and microdissection validation. Brain Structure and Function 2017; 222(4): 1645-62.Jones DK, Christiansen KF, Chapman R, Aggleton JP. Distinct subdivisions of the cingulum bundle revealed by diffusion MRI fibre tracking: implications for neuropsychological investigations. Neuropsychologia 2013; 51(1): 67-78.Kamali A, Flanders AE, Brody J, Hunter JV, Hasan KM. Tracing superior longitudinal fasciculus connectivity in the human brain using high resolution diffusion tensor tractography. Brain Structure and Function 2014; 219(1): 269-81.Latini F, Mårtensson J, Larsson E-M, Fredrikson M, Åhs F, Hjortberg M, Aldskogius H, Ryttlefors M. Segmentation of the inferior longitudinal fasciculus in the human brain: A white matter dissection and diffusion tensor tractography study. Brain Research 2017; 1675: 102-15.Margulies DS, Petrides M. Distinct parietal and temporal connectivity profiles of ventrolateral frontal areas involved in language production. Journal of Neuroscience 2013; 33(42): 16846-52.Petrides M, Pandya DN. Distinct parietal and temporal pathways to the homologues of Broca's area in the monkey. PLoS Biol 2009; 7(8): e1000170.